# Ovarian Decellularized Bioscaffolds Provide an Optimal Microenvironment for Cell Growth and Differentiation In Vitro

**DOI:** 10.3390/cells10082126

**Published:** 2021-08-18

**Authors:** Georgia Pennarossa, Teresina De Iorio, Fulvio Gandolfi, Tiziana A. L. Brevini

**Affiliations:** 1Laboratory of Biomedical Embryology, Department of Health, Animal Science and Food Safety and Center for Stem Cell Research, Università Degli Studi di Milano, 20133 Milan, Italy; georgia.pennarossa@unimi.it (G.P.); teresina.deiorio@unimi.it (T.D.I.); 2Laboratory of Biomedical Embryology, Department of Agricultural and Environmental Sciences—Production, Landscape, Agroenergy, Università Degli Studi di Milano, 20133 Milan, Italy; fulvio.gandolfi@unimi.it

**Keywords:** whole-ovary decellularization, ECM-based scaffold repopulation, epigenetically erased cells, porcine, human, fibroblasts, bioprosthetic ovary, ovarian reconstruction

## Abstract

Ovarian failure is the most common cause of infertility. Although numerous strategies have been proposed, a definitive solution for recovering ovarian functions and restoring fertility is currently unavailable. One innovative alternative may be represented by the development of an “artificial ovary” that could be transplanted in patients for re-establishing reproductive activities. Here, we describe a novel approach for successful repopulation of decellularized ovarian bioscaffolds in vitro. Porcine whole ovaries were subjected to a decellularization protocol that removed the cell compartment, while maintaining the macrostructure and microstructure of the original tissue. The obtained bioscaffolds were then repopulated with porcine ovarian cells or with epigenetically erased porcine and human dermal fibroblasts. The results obtained demonstrated that the decellularized extracellular matrix (ECM)-based scaffold may constitute a suitable niche for ex vivo culture of ovarian cells. Furthermore, it was able to properly drive epigenetically erased cell differentiation, fate, and viability. Overall, the method described represents a powerful tool for the in vitro creation of a bioengineered ovary that may constitute a promising solution for hormone and fertility restoration. In addition, it allows for the creation of a suitable 3D platform with useful applications both in toxicological and transplantation studies.

## 1. Introduction

Premature ovarian failure (POF) is a common endocrine disease that leads to early menopause and infertility. It has an incidence of 0.1% in women under 20 years of age and increases to 1.0–1.5% in subjects younger than 40 years [1,2,3]. Radiotherapy, chemotherapy, viral infections, environmental factors, metabolic and autoimmune disorders, and genetic alterations represent some of the potential causes identified so far. The clinical features displayed by affected patients are hypoestrogenism or estrogen deficiency, increased gonadotropin levels, and amenorrhea [4]. POF also impacts the overall quality-of-life, since alterations in ovarian hormone levels are responsible for different side effects on other organs and tissues, which lead to both medical and psychological problems, such as osteoporosis, cardiovascular disease, and depression [5,6,7]. Although several researchers have been working on the development of different strategies to allow fertility restoration, the prevention and treatment of POF are currently limited. Indeed, to date, several options have been proposed to overcome this challenging issue, including cryopreservation and transplantation of oocytes, embryos, and autologous ovarian fragments, as well as hormone replacement therapies [8,9,10]. However, all of these strategies present limitations and are not fully effective in a complete recovery of the ovarian function [4]. The development of novel and efficient therapeutic alternatives is therefore urgently needed.

In this context, the generation of a bioprosthetic ovary has been proposed as a promising strategy. During the last few years, the “artificial ovary” has received great attention, and many studies have been focusing on the creation of transplantable bioengineered ovaries using different natural polymers and biological materials, such as alginate, collagen, and fibrin three-dimensional matrices [11,12,13,14]. More recently, the extracellular matrix (ECM) has been exploited as the material of choice in the field of tissue engineering and regenerative medicine. ECM-based scaffolds are obtained by decellularization processes, wherein the living cells of an organ are removed, while the ECM components are preserved. This allows for the obtainment of 3D scaffolds that retain the biomechanical and biochemical cues of the native tissue, as well as its ultrastructural architecture, while eliminating immunoreactivity [15]. In contrast to the biological and synthetic matrices previously used, decellularized ECM-based scaffolds are therefore able to recreate in vitro the complex in vivo milieu, promoting the necessary interactions between cells and their surrounding microenvironment, as well as ensuring the correct cell growth, differentiation, and function [2,16,17]. Based on these advantages, decellularization techniques have been applied to a large variety of organs, including the heart [18], lung [19], liver [20], kidney [21], muscle [22], trachea [23], esophagus [24], urinary tissues [25], arteries [26], derma [16], and vagina [27]. 

Whole-ovary decellularization was recently carried out in bovine [28] and porcine species [2,29], leading to the successful production of 3D bioscaffolds for the generation of a bioprosthetic ovary. It must be noted that, to create in vitro an artificial functional organ, we need a recellularization step in which the ECM-based scaffolds are repopulated using suitable cells.

In the present work, we produced porcine whole-ovary decellularized ECM-based scaffolds. We repopulated them with porcine ovarian cells (pOCs) to investigate whether the generated scaffold may constitute a suitable niche for ex vivo culture of ovarian cells. In addition, we engrafted the generated ECM-based matrixes with both porcine and human epigenetically erased cells to explore whether the decellularized scaffolds are able to properly drive cell differentiation, fate, and viability. 

## 2. Materials and Methods

All reagents were purchased from Thermo Fisher Scientific (Milan, Italy) unless otherwise indicated.

### 2.1. Ethical Statement

Human skin biopsies were obtained from healthy female individuals undergoing surgical interventions, after written informed consent and approval of the Ethical Committee of the Ospedale Maggiore Policlinico, Milan, Italy. All methods were carried out in accordance with the approved guidelines.

### 2.2. Collection of Porcine Ovaries

Twenty ovaries were collected from gilts weighing approximately 120 kg at the local slaughterhouse and transported to the laboratory in cold sterile PBS containing 2% antibiotic/antimycotic solution (Sigma, Milan, Italy). Fifteen out of twenty ovaries were subjected to the decellularization protocol for the creation of ovarian ECM-based scaffolds (Figure 1). The remaining 5 ovaries were used as native control tissue. Half of each was immediately fixed in 10% buffered formalin for histological evaluations; the remaining half was subjected to DNA quantification analysis or immersed in RNA later solution (Sigma) for gene expression studies. 

### 2.3. Creation of Decellularized Ovarian ECM-Based Scaffolds

ECM-based scaffolds were obtained by subjecting 15 porcine ovaries to the whole-organ decellularization protocol previously described by Pennarossa et al. [2,29]. Briefly, ovaries were frozen at −80 °C for at least 24 h, thawed at 37 °C in a water bath for 30 min, immersed in 0.5% sodium dodecyl sulfate (SDS; Bio-Rad, Milan, Italy) in deionized water (DI-H_2_O) for 3 h, and incubated overnight in 1% Triton X-100 (Sigma, Milan, Italy) in DI-H_2_O. This step was followed by a wash in DI-H_2_O for 9 h and a subsequent treatment with 2% deoxycholate (Sigma, Milan, Italy) in DI-H_2_O for 12 h. Decellularized whole ovaries were then extensively washed in DI-H2O for 6 h and sterilized with 70% ethanol and 2% antibiotic/antimycotic solution in sterile H_2_O for 30 min. All the steps described were performed using an orbital shaker at 200 rpm. 

Before repopulation, half of each decellularized ovary was subjected to histological analysis, DNA quantification, and the MTT test to evaluate the efficiency of the decellularization process (see Figure 1). The remaining decellularized hemi-ovaries were cut into 9 fragments each of 7 mm in diameter and 1 mm in thickness and repopulated with (A) freshly isolated pOCs, (B) epigenetically erased porcine adult dermal fibroblasts (pEpiE), or (C) epigenetically erased human adult dermal fibroblasts (hEpiE) (Figure 1).

### 2.4. Repopulation of Decellularized Ovarian ECM-Based Scaffolds with pOCs

pOCs were isolated immediately prior to seeding onto decellularized ovarian ECM-based scaffolds.

#### 2.4.1. Collection of Porcine Ovaries and Isolation of pOCs

Five ovaries were collected from gilts weighing approximately 120 kg at the local slaughterhouse and transported to the laboratory in cold sterile PBS containing 2% antibiotic/antimycotic solution (Sigma, Milan, Italy). The ovarian cortex was sliced in ~1 mm^3^ fragments and digested with 1 mg/mL type IV collagenase in HBSS for 45 min, followed by a treatment with 0.25% trypsin/EDTA solution for 15 min. Digested tissues were centrifuged and resuspended in Dulbecco’s Modified Eagle Medium (DMEM) supplemented with 10% fetal bovine serum (FBS), 2 mM glutamine (Sigma, Milan, Italy), and 1% antibiotic/antimycotic solution (Sigma, Milan, Italy).

#### 2.4.2. Repopulation of Decellularized Ovarian ECM-Based Scaffolds

The pOC suspension was dispersed by pipetting, filtered with a 40 μm cell strainer, and seeded onto ECM-based scaffold fragments (7 mm in diameter and 1 mm in thickness). The repopulation process was carried out for 7 days at 37 °C in 5% CO_2_. Half of the medium volume was refreshed every other day. Cultures were arrested after 24 and 48 h and at Day 7, as previously described [2]. Samples were subjected to histological evaluations, DNA quantification, the TUNEL assay, and gene expression analysis.

### 2.5. Repopulation of Decellularized Ovarian ECM-Based Scaffolds with Epigenetically Erased Porcine Adult Dermal Fibroblasts

#### 2.5.1. Collection of Porcine Skin Tissues

Porcine skin tissues were collected from 5 gilts weighing approximately 120 kg at the local slaughterhouse. Specifically, 5 skin specimens of 5 cm^2^ were obtained from an avascular area of the abdomen. Tissues were immersed in sterile PBS containing 2% antibiotic/antimycotic solution (Sigma, Milan, Italy), transported to the laboratory, and used for adult porcine dermal fibroblast isolation. 

#### 2.5.2. Isolation and Culture of Porcine Adult Dermal Fibroblasts

Adult dermal fibroblasts were isolated from fresh skin biopsies obtained from 5 gilts. Tissues were cut into small fragments of ~ 2mm^3^ and transferred into 35 mm^2^ Petri dishes (Sarstedt, Milan, Italy) previously coated with 0.1% gelatin (Sigma, Milan, Italy). Droplets of DMEM supplemented with 20% FBS, 2 mM glutamine (Sigma, Milan, Italy), and 2% antibiotic/antimycotic solution (Sigma, Milan, Italy) were added onto each fragment. Culture dishes were transferred in a 5% CO_2_ incubator at 37 °C in humidified chambers. After 6 days of culture, porcine dermal fibroblasts started to grow out of the original fragments, and the latter were carefully removed. Cells were cultured using the medium described above, grown in 5% CO_2_ at 37 °C, and passaged twice a week at a 1:3 ratio. The porcine primary cell lines obtained from each individual were used in triplicate in 3 independent experiments.

#### 2.5.3. Epigenetic Erasing of Porcine Adult Dermal Fibroblasts with 5-aza-CR and Repopulation of Decellularized Ovarian ECM-Based Scaffolds

Porcine adult dermal fibroblasts were plated onto 0.1% gelatin (Sigma, Milan, Italy) precoated T75 flasks at a concentration of 7.8 × 10^4^ cells/cm^2^. Then, 24 h after seeding, cells were exposed to the epigenetic eraser 5-aza-cytidine (5-aza-CR; Sigma, Milan, Italy) at a 1µM concentration for 18 h. The concentration and time of exposure were selected based on our previous studies [30,31,32,33,34,35,36,37]. At the end of 5-aza-CR exposure, epigenetically erased porcine adult dermal fibroblasts (pEpiE) were used to repopulate the decellularized ovarian ECM-based scaffolds.

To this purpose, 7 × 10^6^ pEpiE were resuspended in 300 μL of DMEM supplemented with 10% FBS, 2 mM glutamine (Sigma, Milan, Italy), and 1% antibiotic/antimycotic solution (Sigma, Milan, Italy) and seeded onto scaffold fragments (7 mm in diameter and 1 mm in thickness), which were then cultured in 5% CO2 at 37 °C for 7 days. Half of the medium volume was refreshed every other day. Cultures were arrested after 24 and 48 h and after 7 days. Samples were subjected to histological evaluations, DNA quantification, the TUNEL assay, and gene expression analysis. pEpiE plated in standard plastic 4-well multidishes (Nunc, Milan, Italy) and cultured in DMEM supplemented with 10% FBS, 2 mM glutamine (Sigma, Milan, Italy), and 1% antibiotic/antimycotic solution (Sigma, Milan, Italy) were used as the control.

### 2.6. Repopulation of Decellularized Ovarian ECM-Based Scaffolds with Epigenetically Erased Human Adult Dermal Fibroblasts

#### 2.6.1. Collection of Human Skin Tissues

Human skin biopsies were collected from 5 healthy women undergoing surgical interventions. Samples were transported to the laboratory in sterile PBS containing 2% antibiotic/antimycotic solution (Sigma, Milan, Italy) and used for adult human dermal fibroblast isolation.

#### 2.6.2. Isolation and Culture of Human Adult Dermal Fibroblasts

Human adult dermal fibroblasts were isolated from skin biopsies freshly collected from 5 healthy women. Samples were cut into ~2 mm^3^ fragments and transferred into 0.1% gelatin (Sigma, Milan, Italy) precoated 35 mm^2^ Petri dishes (Sarstedt, Milan, Italy). Droplets of DMEM supplemented with 20% FBS, 2 mM glutamine (Sigma, Milan, Italy), and 2% antibiotic/antimycotic solution (Sigma, Milan, Italy) were added onto each fragment. Culture was carried out in humidified chambers using a 5% CO_2_ incubator at 37 °C. After 6 days of culture, human dermal fibroblasts grew out of the original tissues, which were carefully removed. Fibroblasts were cultured using the medium described above, grown in 5% CO_2_ at 37 °C and passaged twice a week at a 1:3 ratio. The human primary cell lines obtained from each individual were used in triplicate in 3 independent experiments.

#### 2.6.3. Epigenetic Erasing of Human Adult Dermal Fibroblasts with 5-aza-CR and Repopulation of Decellularized Ovarian ECM-Based Scaffolds

Human fibroblasts were seeded onto T75 flasks previously coated with 0.1% gelatin (Sigma) at a concentration of 7.8 × 10^4^ cells/cm^2^. Then, 24 h after plating, cells were exposed to the epigenetic eraser 5-aza-CR (Sigma) at a 1µM concentration for 18 h [30,31,32,33,34,35,36,37]. At the end of 5-aza-CR exposure, 7 × 10^6^ epigenetically erased human dermal fibroblasts (hEpiE) were resuspended in 300 μL of DMEM supplemented with 10% FBS, 2 mM glutamine (Sigma, Milan, Italy), and 1% antibiotic/antimycotic solution (Sigma, Milan, Italy) and seeded onto scaffold fragments to repopulate the decellularized ovarian ECM-based scaffolds. Cell repopulation was performed in 5% CO_2_ at 37 °C for 7 days. Half of the medium volume was refreshed every other day. Cultures were arrested after 24 and 48 h and at Day 7. Samples were subjected to histological evaluations, DNA quantification, the TUNEL assay, and gene expression analysis. hEpiE plated in standard plastic 4-well multidishes (Nunc, Milan, Italy) and cultured in DMEM supplemented with 10% FBS, 2 mM glutamine (Sigma, Milan, Italy), and 1% antibiotic/antimycotic solution (Sigma, Milan, Italy) were used as the control.

### 2.7. Histological Analysis

Samples were fixed in 10% buffered formalin for 24 h at room temperature, dehydrated in graded alcohols, cleared with xylene, and embedded in paraffin. Serial microtome sections (5 μm thick) were cut, dewaxed, rehydrated, and stained with hematoxylin and eosin (H&E, BioOptica, Milan, Italy) to evaluate the general morphological aspects. ECM components were analyzed with Masson Trichrome staining (Bio-optica, Milan, Italy) for the detection of collagen, Mallory Trichrome staining (Bio-optica, Milan, Italy) for collagen and elastic fibers, Gomori’s aldehyde-fuchsin (Bio-optica) for elastic fibers alone, and Alcian blue (pH 2.5; Bio-optica, Milan, Italy) for the total glycosaminoglycan (GAGs) content. The efficient cell removal was confirmed with 4′,6-diamidino-2-phenylindole (DAPI), and the cell density was measured. Samples were analyzed under an Eclipse E600 microscope (Nikon, Amsterdam, Netherlands) equipped with a digital camera (Nikon, Amsterdam, Netherlands). Pictures were acquired with NIS-Elements Software (Version 4.6; Nikon). Native ovaries were used as the control. 

### 2.8. Stereological Analysis

Volume density (Vv) estimation of collagen, elastin, and GAG was performed on Masson Trichrome, Gomori’s aldehyde-fuchsin, and Alcian blue-stained (pH 2.5) vertical sections, respectively, according to the general Delesse principle. Briefly, the relative volume of each area of interest was estimated from the fractional area of the structure of interest (e.g., collagen) and the total area of the reference compartment (e.g., whole section). Systematic uniform random sampling was applied, and images were captured and superimposed with a point-count stereologic grid with equally distant test points. The points hitting the structure of interest and the respective reference compartment were counted, and the relative volume of each region of interest was then calculated from the respective quotient of points hitting these structures. Vv expressed as percentages was calculated as follows:Vv _(analyzed compartment, reference compartment)_ = [∑P_(analyzed compartment)_/∑P_(reference compartment)_] × 100(1)
where ∑P_(analyzed compartment)_ is the number of points hitting the compartment under study and ∑P_(reference compartment)_ is the number of points hitting the relevant structure.

### 2.9. Cell Density 

The cell number was counted in 5 DAPI-stained tissue sections obtained from each sample. Then, 5 randomly selected fields, at 100× total magnification, were analyzed for each section. Images were acquired with constant exposure parameters and analyzed with ImageJ software version 1.53j (https://imagej.nih.gov/ij/index.html, (accessed on 28 July 2021)), using the Automated Cell Counter tool and following the provider’s instructions. Briefly, 8 bit images were generated, and threshold adjustments were applied. Images were subsequently segmented with a thresholding algorithm in order to highlight areas occupied by the nuclei and to remove the background. Collected data were transformed in binary form. Size and circularity parameters were set, and nuclei were automatically counted. Cell density is expressed per mm2 of tissue.

### 2.10. DNA Quantification 

Ten fragments, ranging from 10 to 24 mg, were cut from each decellularized ovarian ECM-based scaffold and from each repopulated one. Genomic DNA was extracted from each fragment with the PureLink^®^ Genomic DNA Kit, following the manufacturer’s instructions. The DNA concentration was measured with NanoDrop 8000 and normalized against the fragment weights previously annotated. Native ovarian tissues were used as the control. 

### 2.11. TUNEL Assay 

Apoptotic cell death was evaluated with the terminal deoxynucleotidyl transferase-mediated dUTP nick end-labeling (TUNEL) assay. Dewaxed and rehydrated sections were digested with 10 μg/mL proteinase-K (Roche) for 30 min at 37 °C and labelled with the in situ Cell Death Detection Kit, TMR red (Sigma, Milan, Italy), according to the manufacturer’s instructions. Prior to the labelling procedure, positive controls were treated with DNase I recombinant (50 U/mL in 50 mM Tris–HCl, pH 7.5, and 1 mg/mL BSA) for 10 min at 25 °C, to induce DNA strand breaks. For negative controls, TDT was omitted from the reaction mixture. All samples were counterstained with DAPI and mounted with ProLong™ Gold Antifade Mountant. Slides were observed under a Nikon Eclipse 600 microscope. Native tissue was used as the control. TUNEL-positive cells were counted, and their numbers are expressed as a percentage of the total cell counted.

### 2.12. Cytotoxicity Assessment

Decellularized ovarian ECM-based scaffold cytotoxicity was assessed by performing the 3(4,5-dimethylthiazole-2-yl)-2,5-diphenyltetrazolium-bromide (MTT, Roche, Milan, Italy) assay. Briefly, both porcine and human fibroblasts were plated onto 96-well multidishes at a seeding density of 5 × 10^3^ cells/mL. After 24 h, 20 mg of each generated decellularized scaffold was added to cells and cocultured for 1, 3, and 7 days. The, 10 μL of MTT solution was added to the culture media and incubated for 4 h. Formazan salt crystals were dissolved in 100 μL of 10% SDS in 0.01 M HCl overnight. Optical density (OD) was measured at 550 nm. In the control culture (CTR), porcine and human fibroblasts were plated at the same seeding density (5 × 10^3^ cells/mL), and the addition of decellularized whole-ovary fragments was omitted. All experiments were performed in triplicate.

### 2.13. Gene Expression Analysis

RNA was extracted using the TaqManGene Expression Cells to Ct kit (Applied Biosystems) following the manufacturer’s instruction. DNase I was added in lysis solution at a 1:100 concentration. Quantitative PCR was performed on a CFX96 Real-Time PCR (Bio-Rad, Milan, Italy) using predesigned gene-specific primers and probe sets from TaqManGene Expression Assays (see Table 1 for the primer information). ACTB and GAPDH were used as internal reference genes. CFX Manager software 18450000 (Bio-Rad, Milan, Italy) was used for target gene quantification. Gene expression levels are reported with the highest expression set to 1 and the others relative to this.

### 2.14. Statistical Analysis

Statistical analysis was performed using one-way ANOVA (SPSS 19.1; IBM). Data are presented as the mean ± the standard deviation (SD). Differences of *p* ≤ 0.05 were considered significant and are indicated with different superscripts.

## 3. Results

### 3.1. ECM-Based Scaffold Evaluation 

#### 3.1.1. Whole-Organ Decellularization Protocol Eliminates Cellular Components and Maintains Ovarian Macro-Architecture 

Macroscopic observations revealed that, during the decellularization process, the color of the ovaries turned from red to white, while the shape and the homogeneity were well preserved, without any deformation (Figure 2a). 

H & E and DAPI staining showed that the decellularization process was successful and the obtained ECM-based scaffolds were devoid of cells. In particular, basophilic and DAPI staining were absent in decellularized scaffolds (scaffold), while cell nuclei were clearly visible in native ovaries (native), used as the control (Figure 2b). Cell density analysis confirmed a significantly lower number of nuclei in ECM-based scaffolds (scaffold), compared to untreated tissues (native; Figure 2c). In addition, the DNA quantification assay demonstrated a drastic decrease in the DNA content after the whole-organ decellularization process. Specifically, 0.05 ± 0.03 µg DNA/mg of tissue was measured in ECM-based scaffolds (scaffold), while native ovaries contained 1.55 ± 0.21 µg DNA/mg of tissue (native; Figure 2d). 

#### 3.1.2. Whole-Organ Decellularization Protocol Preserves Ovarian ECM Components

Histochemical assessments demonstrated the preservation of the major structural components of the ECM after the whole-organ decellularization process. Both Masson and Mallory Trichrome staining showed the persistence of collagen (blue), which displayed a comparable distribution between ECM-based scaffolds (scaffold) and native tissues (native) (Figure 3a). In addition, Mallory Trichrome staining indicated the maintenance of elastic fibers (red magenta) after the decellularization process (scaffold) (Figure 3a). This was also confirmed by Gomori’s aldehyde-fuchsin staining, demonstrating that the elastic fibers scattered throughout the decellularized ovarian tissues (scaffold), similar to what was detected in native ovaries (native, Figure 3a). Alcian blue staining revealed GAG retention in ECM-based scaffolds (scaffold; Figure 3a). The morphological observations were also confirmed by stereological quantifications, which demonstrated no differences in collagen (Figure 3b), elastin (Figure 3c), and GAGs (Figure 3d) between the ECM-based scaffolds (scaffold) and the native tissue (native).

#### 3.1.3. ECM-Based Scaffolds Show No Cytotoxicity In Vitro

ECM-based scaffold cytocompatibility was determined by the MTT assay, performed using both porcine and human adult dermal fibroblasts. The results obtained demonstrated that the decellularized scaffold exerted no cytotoxic effects. In particular, the OD values detected in porcine (Figure 3e) and human cells (Figure 3f), cocultured with ECM-based scaffolds, were comparable with those obtained in control cultures (CTR), where scaffold fragments were omitted. More in detail, all experimental groups displayed analogous viabilities after 24 and 48 h of culture (Figure 3e,f). In addition, even protracted exposure (seven days) to the decellularized matrix did not induce any cytotoxic response, confirming the efficient removal of the residual chemicals from the ECM-based scaffolds (Figure 3e,f).

### 3.2. pOCs Recellularize the ECM-Based Scaffold In Vitro

Freshly isolated pOCs (Figure 4a) were able to adhere and gradually migrate into the decellularized ECM-based ovarian scaffolds (Figure 4b).

H & E and DAPI staining demonstrated the presence of pOCs in the scaffolds after 24 h of coculture (Figure 4c,d). The histological observations were confirmed by cell density analysis, which showed an increasing number of cells along the culture period (Figure 4e). The TUNEL assay demonstrated the survival of the engrafted cells and showed pOC viability even after seven days of culture (Figure 4g,h). 

In agreement with the morphological data, DNA quantification studies indicated an increase in DNA content. In particular, 0.61 ± 0.05, 1.12 ± 0.1, and 1.34 ± 0.1µg of DNA/mg of tissue were detected after 24 and 48 h and at Day 7 of culture, respectively (Figure 4f).

Gene expression analysis revealed active transcription of the genes specific to the principal active cells of connective tissue, namely VIM and THY1, as well as the markers classically associated with granulosa cells, namely STAR, CYP11A1, CYP19A1, AMH, FSHR, and LHR (Figure 5). Interestingly, the expression levels detected for all the selected genes were comparable between those measured in native tissue (T0) and in repopulated ECM-based scaffolds at all time points considered (Figure 5).

### 3.3. ECM-Based Scaffolds Induce pEpiE and hEpiE Differentiation towards Ovarian Fate 

After epigenetic erasing with 5-aza-CR, porcine and human cells changed their phenotype; the typical fibroblast elongated morphology was replaced by an oval or round shape. Erased cells became smaller with larger nuclei and granular and vacuolated cytoplasm. These changes were functionally accompanied by the onset of the transcription of the pluripotency-related genes OCT4, NANOG, REX1, and SOX2, originally undetectable in untreated fibroblasts (T0; Figure 5. In parallel, both pEpiE and hEpiE significantly downregulated the typical fibroblast markers VIM and THY (Figure 5). 

Both porcine (Figure 6a–c) and human erased cells (Figure 7a–c) were able to repopulate the decellularized ovaries and adhered and gradually migrated into the ECM-based scaffolds. H&E and DAPI staining demonstrated the presence of cells after 24 h of coculture (Figure 6d,e and Figure 7d,e). Cell density analysis showed an increasing number of cells during the period of culture (Figure 6f and Figure 7f). In addition, the TUNEL assay revealed the survival of the engrafted cells and showed pEpiE (Figure 6h,i) and hEpiE (Figure 7h,i) viability even after seven days of culture The morphological data were supported by DNA quantification analysis, indicating an increase in DNA content. In particular, 0.57 ± 0.07, 1.01 ± 0.08, and 1.19 ± 0.09 µg of DNA/mg of tissue were detected after 24 and 48 h and at Day 7 of pEpiE coculture with the ECM-based scaffolds (Figure 6g) and 0.56 ± 0.05, 1.06 ± 0.09, and 1.21 ± 0.11 µg of DNA/mg of tissue at the same time points using hEpiE (Figure 7g). 

Gene expression analysis revealed the onset of the genes specifically associated with granulosa cells, namely STAR, CYP11A1, CYP19A1, AMH, FSHR, and LHR, which were induced, although at low levels, 24 and 48 h postseeding and picked by Day 7 of coculture with levels comparable to those detected in pOC repopulation experiments (Figure 5). Similarly, VIM and THY1 expression values at Day 7 were significantly lower than those identified in untreated fibroblasts (T0), but showed comparable expression levels to those detected when pOCs were used to repopulate. These changes were also paralleled by downregulation of the pluripotency-related genes OCT4, NANOG, REX1, and SOX2. In contrast, when pEpiE and hEpiE were plated onto standard plastic dishes, cells progressively reverted to their original phenotype, and after seven days of culture, the expression of the pluripotency-related genes was completely downregulated, while the VIM and THY1 levels returned comparable to those of untreated fibroblasts (Figure 5). 

## 4. Discussion

Assisted reproduction techniques and hormone replacement therapies presently adopted for the management of POF-affected patients are not fully effective in recovering ovarian functions and do not provide a definitive solution for female fertility restoration. There is therefore an urgent need for novel and efficient therapeutic alternatives, and among the possible strategies, the development of an “artificial ovary” may represent a promising and safe approach to obtain transplantable “structures” that can be used in patients, from childhood to adult age, for restoring endocrine dysfunctions and re-establishing reproductive activities.

In the present study, we described a novel approach to generate porcine decellularized ovarian ECM-based scaffolds and demonstrated their ability to support cell survival and growth and drive cell differentiation. Ovarian ECM-based scaffolds were obtained through the use of a four-step decellularization protocol previously developed in our laboratory. In particular, this involved a freeze–thaw cycle, followed by sequential incubations with SDS, Triton X-100, and deoxycholate, which was previously shown, and here confirmed, for its ability to successfully remove cells, cellular debris, and nuclear material from whole ovaries, while maintaining the macrostructure and microstructure of the original tissue, preserving the shape and homogeneity, and showing no deformation at the end of the process [2,29]. In addition, similar to what was described by Liu et al. [5] and Hassanpour et al. [38], who applied a decellularization protocol to porcine and human ovarian fragments, respectively, the color of the organs turned from red to white, suggesting the ability of the protocol adopted here to induce a significant reduction in cellular components, even when applied to whole ovaries. This was also confirmed by the histological analysis and the DNA quantifications presented in our manuscript, which demonstrated a significant decrement in cell number and DNA content, respectively. 

Parallel with the effective removal of the cellular compartment, a key aspect for the quality of the produced bioscaffolds is the preservation of the ovarian microstructures and its ECM components post-decellularization. The histochemical analysis performed here demonstrated the maintenance of intact collagen and elastic fibers, as well as the persistence of an unaltered distribution of GAGs. This was also confirmed by stereological quantifications, which revealed no differences between native and decellularized ovaries for collagen, elastin, and GAG content, in agreement with what was recently reported by Henning et al., who demonstrated comparable matrisome protein quantities and distributions in porcine decellularized ovaries and the related native tissue [7]. 

Detergent residuals within the decellularized ECM-based scaffolds are also a critical point that needs to be carefully addressed, since this may impair the subsequent scaffold biocompatibility, both in vitro and in vivo, as well as its subsequent recellularization [39]. In this respect, the cytotoxicity assay performed post-decellularization revealed no significant differences between the MTT values detected in porcine and human fibroblasts cocultured with ECM-based scaffolds and those scored in control cultures, even when exposures were protracted up to seven days. To our understanding, this suggests that any toxic effect exerted by the decellularized ovaries is very unlikely.

Altogether, these findings support and expand previous results obtained through the use of decellularization techniques applied to ovarian tissue fragments [5,28] and demonstrate, to our understanding, the possibility to successfully extend the protocol to the whole ovary, leading to the production of ECM-based matrixes suitable for tissue bioengineering.

In this perspective, decellularized ECM-based scaffold quality was tested in repopulation experiments. More in detail, when freshly isolated pOCs were seeded onto the generated biomatrices, cells rapidly adhered and gradually migrated into the bioscaffolds within the first 24 h. The cell homing ability was confirmed both by H&E and DAPI staining, as well as by cell density analysis, which demonstrated an increase in cell number along the culture period. In parallel, the TUNEL assay showed the repopulating cell viability as being stably maintained in culture, with apoptotic indices comparable to those scored in the native tissue. These observations suggested the decellularized ECM-based scaffolds’ ability to support ovarian cell survival in vitro for a protracted culture period, as demonstrated by DNA quantification studies, which showed an increase in DNA content with 0.61 ± 0.05, 1.12 ± 0.1, and 1.34 ± 0.1 µg of DNA/mg of tissue after 24 h and 48 h and at Day 7 of culture, respectively. These data are in full agreement and further expand previous studies reporting the ability of decellularized ovarian fragments to successfully support murine and human ovarian cell engraftment in vitro [5,17,38]. In addition, they demonstrated the possibility to use whole-ovary decellularized ECM-based scaffolds as suitable niches for pOC ex vivo culture. 

Molecular evaluations of engrafted pOCs demonstrated active and steady transcription for the main ovary-specific genes. In particular, we could detect the expression of genes specifically associated with granulosa cells, namely STAR, CYP11A1, CYP19A1, AMH, FSHR, and LHR, as well as connective tissue markers, such as VIM and THY1. Interestingly, the results obtained indicated that the relative expression levels of each specific gene vs. all the other genes considered was maintained for all of the engrafting period. Although further studies are needed to better elucidate this aspect, it is tempting to speculate that, different from what was reported for 2D culture systems where a poor correlation with in vivo conditions was detected [40,41], the decellularized ECM-based scaffold niche may provide repopulating cells with inputs that preserve the transcription regulatory mechanisms at work in their native tissue, offering an adequate 3D microenvironment driving repopulating cells. This hypothesis found further support in the results obtained when ECM-based scaffolds were repopulated with epigenetically erased cells obtained through epigenetic erasing [30,31,32,33,34,35,36,37,42,43,44,45,46,47]. Similar to what was observed when pOCs were used, both pEpiE and hEpiE rapidly adhered and migrated into the bioscaffolds, suggesting the ability of the decellularized ECM-based scaffolds to provide a suitable environment and support epigenetically erased cell attachment, migration, and growth. Notably, gene expression analysis showed striking changes in the transcription pattern, with downregulation of the pluripotency-related genes OCT4, NANOG, REX1, and SOX2 and the onset of the main genes specifically associated with granulosa cells, namely STAR, CYP11A1, CYP19A1, AMH, FSHR, and LHR, which were originally undetectable in repopulating cells. In particular, the expression levels of the above-mentioned genes, which have been commonly used for tracking ESC differentiation into mature and functional granulosa cells [48,49], were detectable after 24 and 48 h and reached values comparable with those detected in pOC repopulation experiments at Day 7 of culture, indicating ECM-based scaffolds’ ability to properly address and drive erased cell differentiation in vitro.

## 5. Conclusions

The results described in this manuscript demonstrated that decellularized ECM-based scaffolds may provide an optimal 3D microenvironment for ex vivo culture of ovarian cells, supporting the maintenance of the original expression pattern. Furthermore, they are able to properly drive epigenetically erased cell differentiation, fate, and viability in vitro, encouraging repopulating cells to adopt a transcriptional shift in response to the scaffold environment. Overall, the strategy proposed here allows for the creation of a suitable 3D platform to investigate solutions for hormone and fertility restoration, toxicological and drug testing, as well as transplantation studies.

## Figures and Tables

**Figure 1 cells-10-02126-f001:**
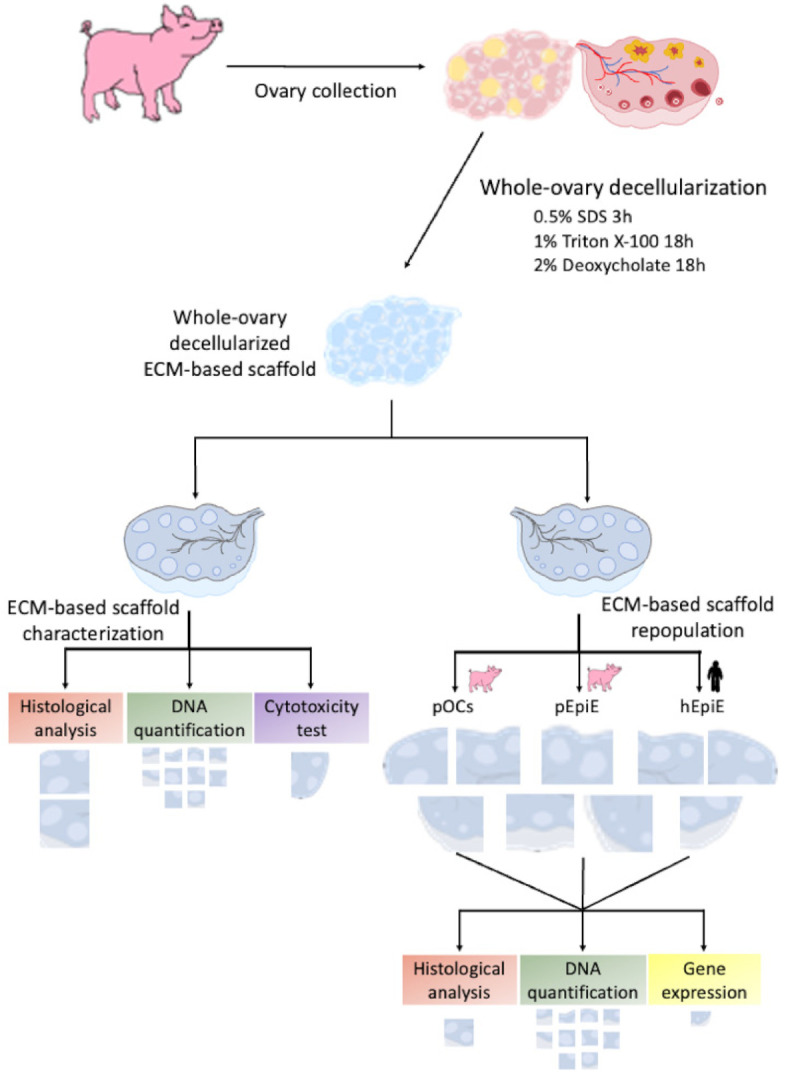
Scheme showing overall experimental design.

**Figure 2 cells-10-02126-f002:**
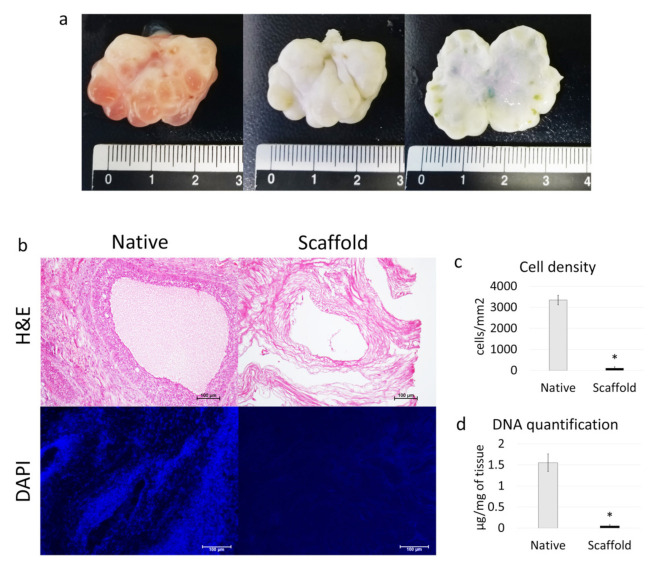
Macroscopic and microscopic evaluations of ECM-based scaffolds and DNA quantification. (**a**) Native (left panel) and decellularized (middle and right panels) ovaries display comparable shapes and homogeneity, while the color turns from red (left panel) to white (middle and right panels); (**b**) H & E staining shows the presence of both basophilic (cell nuclei) and eosinophilic (cell cytoplasm and ECM) staining in the control tissue (native), while cell nuclei and the related basophilic staining are absent in the decellularized ECM-based scaffolds (scaffold). DAPI staining displays the presence of nuclei in native ovaries (native), which disappeared after the decellularization process (scaffold). Scale bars = 100 μm; (**c**) Cell density demonstrated a significantly lower number of nuclei in the decellularized ECM-based scaffolds (scaffold) compared to the untreated tissues (native); data are expressed as the mean ± the standard error of the mean (SEM), * *p* < 0.05; (**d**) DNA quantification analysis showed a significant decrease in the DNA content of the decellularized ECM-based scaffolds (scaffold) compared to the native tissue (native). Data are expressed as the mean ± the standard error of the mean (SEM), * *p* < 0.05.

**Figure 3 cells-10-02126-f003:**
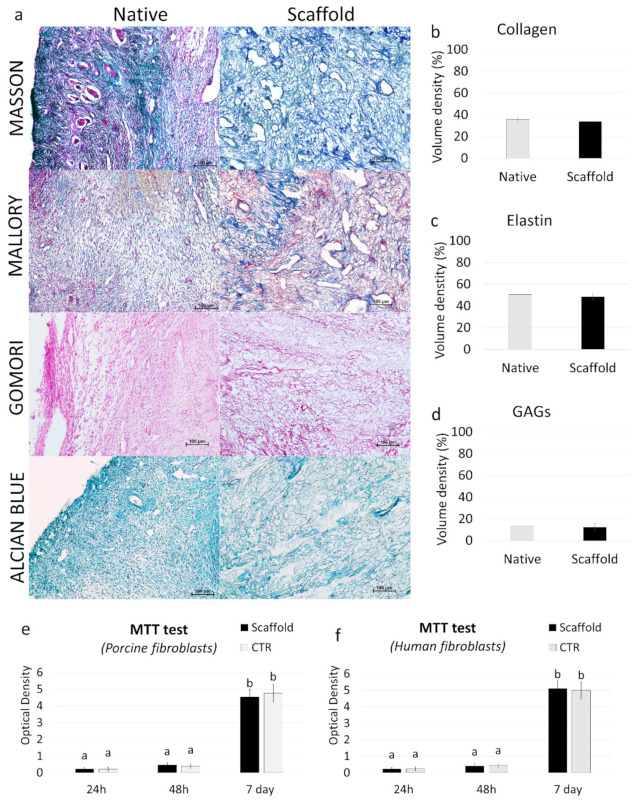
ECM microarchitecture and composition in ECM-based scaffolds and cytotoxicity assessment. (**a**) Masson’s Trichrome staining showed the persistence of collagen fibers (blue) and their comparable distribution between the native ovaries (native) and the decellularized tissues (scaffold). Mallory’s Trichrome staining demonstrated the maintenance of intact collagen (blue) and elastic fibers (red magenta) after the decellularization process (scaffold). Gomori’s aldehyde-fuchsin staining confirmed that ECM-based scaffolds (scaffold) retained the elastic fibers scattered throughout the decellularized ovary, similar to what was visible in untreated ovaries (native). Alcian blue staining revealed GAG retention in the decellularized scaffolds (scaffold). Scale bars = 100 μm. (**b–d**) Stereological quantifications demonstrated no significant differences between untreated ovaries (native) and the decellularized ECM-based scaffolds (scaffold) in collagen (**b**), elastin (**c**), and GAG (**d**) contents. Data are expressed as the mean ± the standard error of the mean (SEM) (*p* > 0.05). (**e–f**) MTT assays demonstrated no significant differences in the OD values of porcine (**e**) and human (**f**) adult dermal fibroblasts cocultured with ECM-based scaffold fragments and those identified in control cultures (CTR) at all the different time points analyzed. Data are expressed as the mean ± the standard error of the mean (SEM) (*p* > 0.05). a and b indicate statistically significant differences (*p* < 0.05).

**Figure 4 cells-10-02126-f004:**
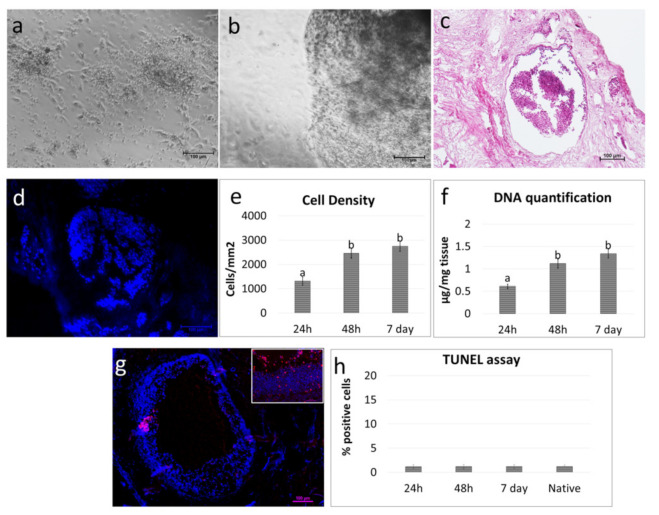
Repopulation of decellularized ovarian ECM-based scaffolds with pOCs. (**a**) Image illustrating freshly isolated pOCs. Scale bar = 100 μm. (**b**) pOCs adhered and migrated into the decellularized ECM-based ovarian scaffolds. Representative image after 7 days of culture. Scale bar = 100 μm. (**c**) H & E staining demonstrated the presence of pOCs in the ECM-based scaffolds. Scale bar = 100 μm. (**d**) DAPI staining confirmed the positivity for nuclei. Scale bar = 100 μm. (**e**) Cell density analysis showed ECM-based scaffolds’ repopulation after 24 h, with a higher number of cells after 48 h and 7 days of coculture. Data are expressed as the mean ± the standard error of the mean (SEM). a and b indicate statistically significant differences (*p* < 0.05). (**f**) DNA quantification demonstrated the presence of pOCs into the ECM-based scaffolds after 24 h, with an increasing cell number at 48 h, which was steadily maintained at Day 7. Data are expressed as the mean ± the standard error of the mean (SEM). a and b indicate statistically significant differences (*p* < 0.05). (**g**) Representative picture of the TUNEL assay showing positive cells in red. Nuclei were stained with DAPI (blue). Scale bars = 100 μm and 50 μm. (**h**) TUNEL-positive cell rates detected 24–48 hours and 7 days postseeding were comparable to those identified in the native tissues (native). Data are expressed as the mean ± the standard error of the mean (SEM).

**Figure 5 cells-10-02126-f005:**
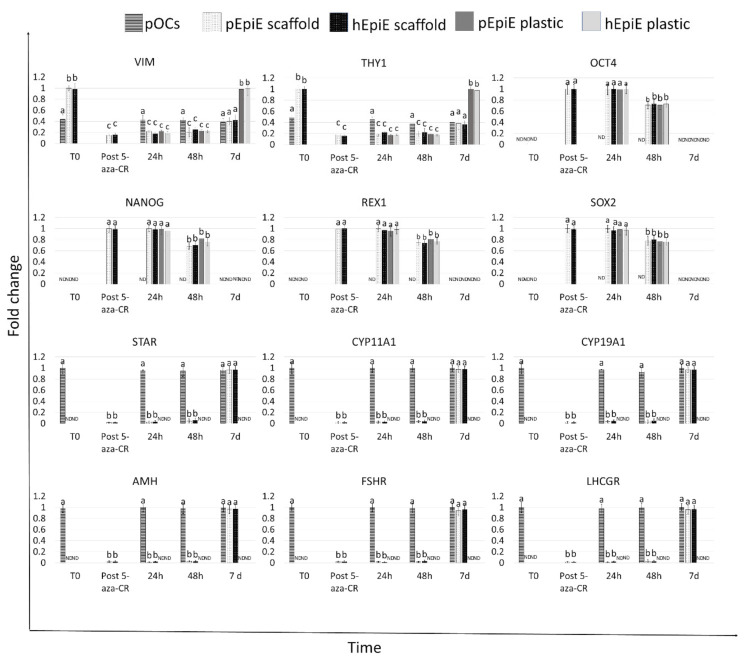
Gene expression changes in pOCs (line bars), pEpiE (white with black dots bars), and hEpiE (black with white dots bars) during the ECM-based scaffold repopulation process. Expression pattern of fibroblast-specific (VIM, and THY1), pluripotency-related (OCT4, NANONG, REX1, and SOX2), and granulosa-cell-associated (STAR, CYP11A1, CYP19A1, AMH, FSHR, and LHCGR) markers in freshly isolated pOCs and porcine and human untreated fibroblasts (T0), in fibroblasts exposed to 5-aza-CR (post 5- aza-CR), and at different time points of ECM-based scaffold repopulation (24 and 48 h and 7 days). (a), (b), and (c) indicate statistically significant differences (*p* < 0.05).

**Figure 6 cells-10-02126-f006:**
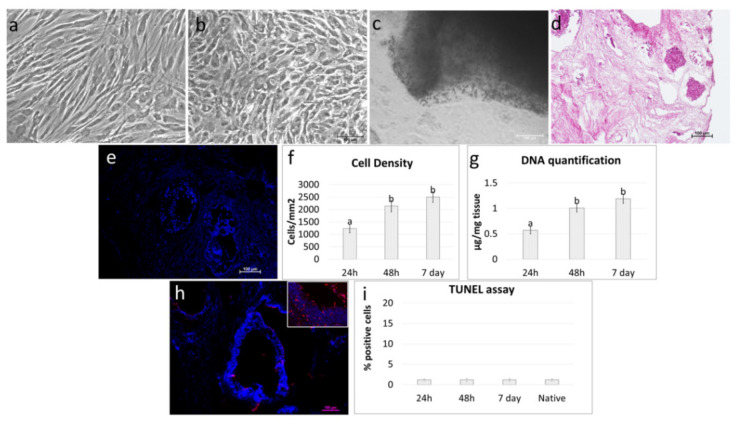
Repopulation of decellularized ovarian ECM-based scaffolds with pEpiE. (**a**) Representative image of porcine untreated fibroblasts showing the typical elongated shape. Scale bar = 100 μm. (**b**) After 18 h of exposure to 5-aza-CR, the cells displayed a round epithelioid aspect and became smaller in size with larger nuclei and granular cytoplasm. Scale bar = 100 μm. (**c**) pEpiE adhered and migrated to the decellularized ECM-based ovarian scaffolds. Representative image after 7 days of culture. Scale bar = 100 μm. (**d**) H & E staining demonstrated the presence of pEpiE in the ECM-based scaffolds. Scale bar = 100 μm. (**e**) DAPI staining confirmed the positivity for nuclei. Scale bar = 100 μm. (**f**) Cell density analysis showed ECM-based scaffolds’ repopulation after 24 h, with a higher number of cells after 48 h and 7 days of coculture. Data are expressed as the mean ± the standard error of the mean (SEM). a and b indicate statistically significant differences (*p* < 0.05). (**g**) DNA quantification demonstrated the presence of pEpiE in the ECM-based scaffolds after 24 h, with an increasing number of cells at 48 h, which was steadily maintained at Day 7. Data are expressed as the mean ± the standard error of the mean (SEM). a, and b indicate statistically significant differences (*p* < 0.05). (**h**) Representative picture of the TUNEL assay showing positive cells in red. Nuclei were stained with DAPI (blue). Scale bars = 100 μm and 50 μm. (**i**) TUNEL-positive cell rates detected 24–48 h and 7 days after seeding were comparable to those identified in the native tissues (native). Data are expressed as the mean ± the standard error of the mean (SEM).

**Figure 7 cells-10-02126-f007:**
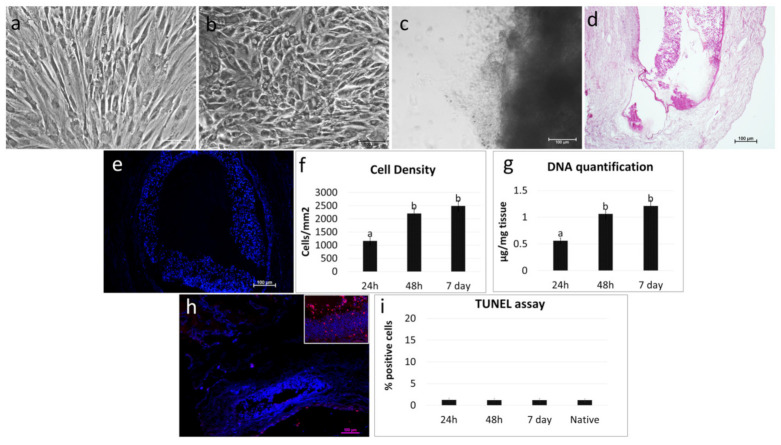
Repopulation of decellularized ovarian ECM-based scaffolds with hEpiE. (**a**) Representative image of human untreated fibroblasts showing the typical elongated shape. Scale bar = 100 μm. (**b**) After 18 h of exposure to 5-aza-CR, cells displayed a round epithelioid aspect and became smaller in size with larger nuclei and granular cytoplasm. Scale bar = 100 μm. (**c**) hEpiE adhered and migrated into the decellularized ECM-based ovarian scaffolds. Representative image after 7 days of culture. Scale bar = 100 μm. (**d**) H & E staining demonstrated the presence of hEpiE in the ECM-based scaffolds. Scale bar = 100 μm. (**e**) DAPI staining showed the positivity for nuclei. Scale bar = 100 μm. (**f**) Cell density analysis indicated ECM-based scaffolds’ repopulation after 24 h, with a higher number of cells after 48h and 7 days of coculture. Data are expressed as the mean ± the standard error of the mean (SEM). a and b indicate statistically significant differences (*p* < 0.05). (**g**) DNA quantification confirmed the presence of hEpiE in the ECM-based scaffolds after 24 h, with an increasing number of cells at 48h, which was steadily maintained at Day 7. Data are expressed as the mean ± the standard error of the mean (SEM). a and b indicate statistically significant differences (*p* < 0.05). (**h**) Representative picture of the TUNEL assay showing positive cells in red. Nuclei were stained with DAPI (blue). Scale bars = 100 μm and 50 μm. (**i**) TUNEL-positive cell rates detected 24–48 hours and 7 days after seeding were comparable to those identified in the native tissues (native). Data are expressed as the mean ± the standard error of the mean (SEM).

**Table 1 cells-10-02126-t001:** List of primers used for quantitative PCR analysis.

Gene	Species	Description	Cat. N.
ACTB	Sus scrofa	Actin, beta	Ss03376563_uH
ACTB	Homo sapiens	Actin, beta	Hs99999903_m1
AMH	Sus scrofa	Anti-Mullerian hormone	Ss03383931_u1
AMH	Homo sapiens	Anti-Mullerian hormone	Hs00174915_m1
CYP11A1	Sus scrofa	Cytochrome P450 family 11 subfamily A member 1	Ss03384849_u1
CYP11A1	Homo sapiens	Cytochrome P450 family 11 subfamily A member 1	Hs00167984_m1
CYP19A1	Sus scrofa	Cytochrome P450 family 19 subfamily a member 1	Ss03384876_u1
CYP19A1	Homo sapiens	Cytochrome P450 family 19 subfamily a member 1	Hs00903411_m1
FSHR	Sus scrofa	Follicle-stimulating hormone receptor	Ss03384581_u1
FSHR	Homo sapiens	Follicle-stimulating hormone receptor	Hs01019695_m1
GAPDH	Sus scrofa	Glyceraldehyde-3-phosphate dehydrogenase	Ss03375629_u1
GAPDH	Homo sapiens	Glyceraldehyde-3-phosphate dehydrogenase	Hs02786624_g1
LHCGR	Sus scrofa	Luteinizing hormone/choriogonadotropin receptor	Ss03384991_u1
LHCGR	Homo sapiens	Luteinizing hormone/choriogonadotropin receptor	Hs00174885_m1
NANOG	Sus scrofa	Nanog homeobox	Ss04245375_s1
NANOG	Homo sapiens	Nanog homeobox	Hs02387400_g1
OCT4	Sus scrofa	POU Class 5 Homeobox 1	Ss03389800_m1
OCT4	Homo sapiens	POU Class 5 Homeobox 1	Hs00999632_g1
REX1	Sus scrofa	ZFP42 zinc finger protein	Ss03373622_g1
REX1	Homo sapiens	ZFP42 zinc finger protein	Hs01938187_s1
SOX2	Sus scrofa	SRY-Box Transcription Factor 2	Ss03388002_u1
SOX2	Homo sapiens	SRY-Box Transcription Factor 2	Hs04234836_s1
STAR	Sus scrofa	Steroidogenic acute regulatory protein	Ss03381250_u1
STAR	Homo sapiens	Steroidogenic acute regulatory protein	Hs00986559_g1
THY1	Sus scrofa	Thy-1 cell surface antigen	Ss03376963_u1
THY1	Homo sapiens	Thy-1 cell surface antigen	Hs00174816_m1
VIM	Sus scrofa	Vimentin	Ss04330801_gH
VIM	Homo sapiens	Vimentin	Hs05024057_m1

## Data Availability

The data presented in this study are available on request from the corresponding author.

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
