# Peer review of "Ovarian Decellularized Bioscaffolds Provide an Optimal Microenvironment for Cell Growth and Differentiation In Vitro"

_cells, 2021, doi:10.3390/cells10082126_

Round 1

Reviewer 1 Report

In this manuscript, the authors prepared decellularized ovarian tissues and cultured the cells in/on the decellularized ECM. The decellularized ECM can support cell adhesion and growth. Moreover, the decellularized ECM can promote the differentiation toward the ovarian fate. Overall, the manuscript is well-written. However, the results did not support the conclusion in part. So, I suggested to revise the manuscript for the publication. Specific comments are below.

1) In Figure 4h, the authors performed TUNEL assay. Apoptotic cells were detected even in native tissues. Why? Is this apoptosis due to the damage during tissue isolation?

2) In Figure 5, the authors demonstrated gene expression in the cells cultured on/in the decellularized ECM. The authors claimed that the decellularized ECM induce the differentiation towards the ovarian fate. But it lacks experimental controls (e.g., the culture on tissue culture plates). I was wondering whether the cells recovered ovarian phenotype spontaneously. The authors should show the results of experimental control.

3) There are several typos (e.g., “tunnel assay” in line 266, “DNAse” in line 272). The authors should check again through the manuscript.

4) The authors should describe the condition of each experiment. For example, what is the control culture in Figure 3? How long did the authors culture the cells in Figure 4b, 6c, and 7c?

Author Response

In this manuscript, the authors prepared decellularized ovarian tissues and cultured the cells in/on the decellularized ECM. The decellularized ECM can support cell adhesion and growth. Moreover, the decellularized ECM can promote the differentiation toward the ovarian fate. Overall, the manuscript is well-written. However, the results did not support the conclusion in part. So, I suggested to revise the manuscript for the publication. Specific comments are below.

We thank the Reviewer for the comments and suggestions that will help improving the quality of our manuscript.

1) In Figure 4h, the authors performed TUNEL assay. Apoptotic cells were detected even in native tissues. Why? Is this apoptosis due to the damage during tissue isolation?

We thank the Reviewer for raising this point and would like to add a general comment about apoptosis, which is reported to be present and essential for the maintenance of homeostasis in all tissues, including the ovary (Andreu-Vieyra CV and Habibi HR. 2000. Factors controlling ovarian apoptosis. Can J Physiol Pharmacol; 78: 1003-12, Tilly JL. 1996. Apoptosis and ovarian function. Rev Reprod;1: 162-72.). In particular, the first morphological observations demonstrating apoptosis in the ovarian tissue dates back to 1885 (Clarke PG, Clarke S 1996 Nineteenth century research on naturally occuring cell death and related phenomena. Anatomy and Embryology (Berlin) 193, 81–99). Subsequently numerous studies confirmed the presence of cell death processes in the ovary and elucidated that their regulation depends on a balance between anti- and pro-apoptotic factors (Guthrie HD, et al. 1998. Follicle stimulating hormone and insulin-like growth factor-I attenuate apoptosis in cultured porcine granulosa cells. Biol, Reprod; 58: 390-6. 4. Rouillier P, et al. 1998. Steroid production, cell proliferation, and apoptosis in cultured bovine antral and mural granulosa cells: development of an in vitro model to study estradiol production. Mol Reprod Dev; 50: 170-7.). At the same time, we would also like to highlight that the apoptotic values reported in the present manuscript for both native tissues and repopulated scaffolds are comparable to those commonly detected in fresh ovarian tissues (Sandra Sanfilippo,et al. 2013. Quality and functionality of human ovarian tissue after cryopreservation using an original slow freezing procedure. J Assist Reprod Genet.; 30(1): 25–34. 2012. doi: 10.1007/s10815-012-9917-5; Peter R. Hurst, et al. 2006. Caspase-3, TUNEL and ultrastructural studies of small follicles in adult human ovarian biopsies. Human Reproduction, 21, 8, 1974–1980; C Jiaojiao  et al.  2021. Long-time low-temperature transportation of human ovarian tissue before cryopreservation. Reproductive BioMedicine Online 43, 2, 172-183)

2) In Figure 5, the authors demonstrated gene expression in the cells cultured on/the decellularized ECM. The authors claimed that the decellularized ECM induce the differentiation towards the ovarian fate. But it lacks experimental controls (e.g., the culture on tissue culture plates). I was wondering whether the cells recovered ovarian phenotype spontaneously. The authors should show the results of experimental control.

This is indeed an interesting observation, and we thank the Reviewer for that. We did not include the results obtained in the experimental controls because we did not want to be redundant. Indeed, we know and published that both porcine and human epigenetically erased cells seeded onto standard plastic plates spontaneously revert to their original phenotype. In the present experiments,  the behavior was the same and within 7 days of culture, they completely down-regulate the expression of pluripotency-related genes, while displaying transcription levels for fibroblast specific marker VIM and THY1 comparable to those identified in untreated cells.This  in line with data previously obtained in our laboratory demonstrating that fibroblasts exposed to 5-aza-CR, in the absence of other inductive stimuli, progressively reacquired the phenotype distinctive of the starting cell population (Pennarossa et al. Brief demethylation step allows the conversion of adult human skin fibroblasts into insulin-secreting cells. Proc Natl Acad Sci U S A. 2013, Pennarossa et al. Reprogramming of Pig Dermal Fibroblast into Insulin Secreting Cells by a Brief Exposure to 5-aza-cytidine. SCTR 2014; Brevini et al., Epigenetic conversion of adult dog skin fibroblasts into insulin-secreting cells. Vet J. 2016; Manzoni et al. 5-azacytidine affects TET2 and histone transcription and reshapes morphology of human skin fibroblasts. Sci Rep 2016; Pennarossa et al. Use of a PTFE Micro-Bioreactor to Promote 3D Cell Rearrangement and Maintain High Plasticity in Epigenetically Erased Fibroblasts. STCR 2019; Pennarossa et al. A Two-Step Strategy that Combines Epigenetic Modification and Biomechanical Cues to Generate Mammalian Pluripotent Cells. JoVE 2020).

As requested by the Reviewer, however, details describing the experimental controls were now added in the text and in Figure 5.

Please see:

  • M&M section, lines 183-185: “pEpiE plated in standard plastic 4-well multidish (Nunc) and cultured in DMEM supplemented with 10% FBS, 2 mM glutamine (Sigma), and 1% antibiotic/antimycotic solution (Sigma) were used as control.”
  • M&M section, lines 202-222: “hEpiE plated in standard plastic 4-well multidish (Nunc) and cultured in DMEM supplemented with 10% FBS, 2 mM glutamine (Sigma), and 1% antibiotic/antimycotic solution (Sigma) were used as control.”
  • Results section, lines 478-481: “In contrast, when pEpiE and hEpiE were plated onto standard plastic dishes, cells progressively reverted to their original phenotype and, after 7 days of culture, the expression of the pluripotency-related genes was completely down-regulated and the VIM and THY1 levels returned comparable to those of untreated fibroblasts (Figure 5).”

3) There are several typos (e.g., “tunnel assay” in line 266, “DNAse” in line 272). The authors should check again through the manuscript.

We apologise for this. The manuscript was carefully checked

4) The authors should describe the condition of each experiment. For example, what is the control culture in Figure 3? How long did the authors culture the cells in Figure 4b, 6c, and 7c?

As requested by the Reviewer the conditions of each experiment were included in the revised version of the manuscript in the Material and Methods section.

M&M section for Figure 3: Lines 309-311 “In the control culture (CTR), porcine and human fibroblasts were plated at the same seeding density (5 × 103 cells/ml) and the addition of decellularized whole-ovary fragments was omitted.”

M&M section for Figure 4b: Line 142 “Repopulation processes was carried for 7 days at 37°C in 5% CO2” and line 422 “Representative imagine after 7 days of culture.”

M&M section for Figure 6c: Line 180 “…that were then cultured in 5% CO2 at 37°C for 7 days.” and line 488 “Representative imagine after 7 days of culture.”

M&M section for Figure 7c: Line 217 “Cell repopulation was performed in 5% CO2 at 37°C for 7 days.” and line 507 “Representative imagine after 7 days of culture.”

Reviewer 2 Report

In this manuscript, the authors present their work on decelluarizing porcine ovaries and repopulating porcine ovarian cells. This work demonstrates the feasibility of using decellularized bioscaffolds that can be potentially used as “artificial ovaries.” The authors showed that porcine ovary cells and epigenetically erased porcine dermal fibroblasts and human fibroblasts could be repopulated in the decellularized porcine ovarian scaffold without inducing toxicity. It is a well-written manuscript that includes thorough characterizations of the viability of the decellularized scaffolds.

Here are minor suggestions and comments:

-For figure presentations, it will be easier to identify figures if the labels were placed in the upper left corner of each figure rather than the lower right corner.

-In Figures 3e-f, Figures 4e-f, Figure 5, Figure 6f-g, Figure 7f-g, “a” and “b” are shown, please define what they mean.

-3.1.3. Page 12. High OD values after 7 days: does it mean that cells proliferated which then increased the total metabolites?

-For gene expression analyses (page 13, lines 407-409), could the authors elaborate on why those genes were selected and what those selected genes are important in the study, perhaps in the Discussion section?

-Will human ovarian cells have the same positive outcome when seeded with the porcine ovarian scaffold?

Author Response

In this manuscript, the authors present their work on decelluarizing porcine ovaries and repopulating porcine ovarian cells. This work demonstrates the feasibility of using decellularized bioscaffolds that can be potentially used as “artificial ovaries.” The authors showed that porcine ovary cells and epigenetically erased porcine dermal fibroblasts and human fibroblasts could be repopulated in the decellularized porcine ovarian scaffold without inducing toxicity. It is a well-written manuscript that includes thorough characterizations of the viability of the decellularized scaffolds.

We thank the Reviewer for appreciating our manuscript and for the comments and suggestions that will greatly improve the quality of the text.

Here are minor suggestions and comments:

-For figure presentations, it will be easier to identify figures if the labels were placed in the upper left corner of each figure rather than the lower right corner.

As suggested by the Reviewer, figure labels were moved in the upper left corner of each figure.

-In Figures 3e-f, Figures 4e-f, Figure 5, Figure 6f-g, Figure 7f-g, “a” and “b” are shown, please define what they mean.

 We apologize for this. “a” and “b” are now defined for their meaning in all Figures.

-3.1.3. Page 12. High OD values after 7 days: does it mean that cells proliferated which then increased the total metabolites?

The high OD values detected after 7 days indicate fibroblast proliferation that results in total metabolite increment.

-For gene expression analyses (page 13, lines 407-409), could the authors elaborate on why those genes were selected and what those selected genes are important in the study, perhaps in the Discussion section?

As suggested by the Reviewer, a sentence was added in the Discussion section, please see lines 606-611.

“In particular, the expression levels of the above mentioned genes, that were commonly used for tracking ESC differentiation into mature and functional granulosa cells [49,50], were detectable after 24 and 48 hours and reached values comparable with those detected in pOC repopulation experiments at day 7 of culture, indicating ECM-based scaffold ability to properly address and drive erased cell differentiation in vitro.”

-Will human ovarian cells have the same positive outcome when seeded with the porcine ovarian scaffold?

The results described in the present manuscript demonstrate that decellularized ECM-based scaffolds provide an optimal 3D microenvironment for ex vivo culture of porcine ovarian cells. In parallel, these scaffolds appear to be able to encourage and drive both porcine and human high plasticity cell differentiation towards ovarian fate. This leads us to speculate that the decellularized ovarian scaffolds may maintain distinctive bio-chemical and bio-mechanical cues specific of the native tissue that can be recognized from both porcine and human cells, driving them in a tissue-specific way. Although further studies are needed, we may hypothesize that the decellularized ECM-based scaffolds produced in the experiments here described may also provide a suitable niche for the ex vivo culture of ovarian cells of human origin.

Round 2

Reviewer 1 Report

The authors properly answered to the reviewer's comments. However, I still have a question. The authors claimed that ECM scaffolds "induced (or drived)" cell differentiation. On the other hand, the cells on plastic dish also differentiated spontaneously and the differentiation levels were comparable to the cells on ECM scaffolds. It seemed that ECM scaffolds did not inhibit cellular spontaneous differentiation. I was wondering how the authors could claim that ECM "induced" diffferentiation. I felt that this is overinterpretion.

Author Response

The authors properly answered to the reviewer's comments. However, I still have a question. The authors claimed that ECM scaffolds "induced (or drived)" cell differentiation. On the other hand, the cells on plastic dish also differentiated spontaneously and the differentiation levels were comparable to the cells on ECM scaffolds. It seemed that ECM scaffolds did not inhibit cellular spontaneous differentiation. I was wondering how the authors could claim that ECM "induced" diffferentiation. I felt that this is overinterpretion.

Once again, we thank the Reviewer for giving us the opportunity to clarify this point.

From these and previous experiments (Pennarossa et al. Brief demethylation step allows the conversion of adult human skin fibroblasts into insulin-secreting cells. Proc Natl Acad Sci U S A. 2013, Pennarossa et al. Reprogramming of Pig Dermal Fibroblast into Insulin Secreting Cells by a Brief Exposure to 5-aza-cytidine. SCTR 2014; Brevini et al., Epigenetic conversion of adult dog skin fibroblasts into insulin-secreting cells. Vet J. 2016; Manzoni et al. 5-azacytidine affects TET2 and histone transcription and reshapes morphology of human skin fibroblasts. Sci Rep 2016; Pennarossa et al. Use of a PTFE Micro-Bioreactor to Promote 3D Cell Rearrangement and Maintain High Plasticity in Epigenetically Erased Fibroblasts. STCR 2019; Pennarossa et al. A Two-Step Strategy that Combines Epigenetic Modification and Biomechanical Cues to Generate Mammalian Pluripotent Cells. JoVE 2020),  we know that exposure to 5-aza-CR drives fibroblasts in a high permissive state. This state is transient and reversible. If we seed these cells on plastic, they loose the acquired permissive state and revert spontaneously to their original phenotype, eg. fibroblasts. Of course, in alternative, taking advantage of the highly permissive window, we can also readdress cells towards a different phenotype. We can achieve this by using either a proper differentiation medium and/or a proper 3D matrix with the correct stiffness.

In this work, we apply the same reasoning and test whether the use of  a proper 3D matrix (the ovarian ECM-based scaffold)  is able to encourage 5-aza-CR treated fibroblasts towards the ovarian phenotype. Although, we clearly state that further experiments are needed, we think that the results presented are very encouraging and indicate a contribution from the scaffold. Indeed, when the same cells were seeded on plastic, they  reverted to their original  phenotype (fibroblasts) and did not give any evident sign of transition towards the ovarian phenotype (please see lines 478-482 and figure 5: pEpiE plastic and hEpiE plastic, 7 d).